# Honeydew Is a Food Source and a Contact Kairomone for *Aphelinus mali*

**DOI:** 10.3390/insects14050426

**Published:** 2023-04-29

**Authors:** Ainara Peñalver-Cruz, Pascale Satour, Bruno Jaloux, Blas Lavandero

**Affiliations:** 1Instituto de Investigación y Tecnología Agroalimentaria (IRTA)-Campus de la Escola Tècnica Superior d’Enginyeria Agrària (ETSEA), Protecció Vegetal Sostenible, Av. Rovira Roure 191, 25198 Lleida, Spain; 2IRHS, INRAE, Institut Agro, Université d’Angers, 49071 Beaucouzé, France; pascale.satour@univ-angers.fr; 3IGEPP, INRAE, Institut Agro, Université de Rennes, 49000 Angers, France; bruno.jaloux@institut-agro.fr; 4Laboratorio de Control Biológico, Instituto de Ciencias Biológicas, Universidad de Talca, Talca 3465548, Chile; blavandero@utalca.cl

**Keywords:** parasitoids, sugars, biological control, *Eriosoma lanigerum*, chemical cues

## Abstract

**Simple Summary:**

Natural enemies of major pests such as parasitoids require sugar-rich food for development and reproduction. In agricultural fields, honeydew excreted by aphids is often the predominant sugar source that parasitoids can consume. Moreover, honeydew can constitute a cue used by parasitoid females to find their aphid host. However, in some species, such as the woolly apple aphid (*Eriosoma lanigerum*), honeydew is coated with a thin layer of wax, which could make its consumption harder and prevent the emission of attractive odors for parasitoids. In the present study, we evaluated the benefits in terms of longevity and host searching that could provide honeydew to the main parasitoid of the woolly apple aphid, *Aphelinus mali*, and infer parasitoid feeding patterns in apple orchards. Results suggested that *A. mali* is able to consume honeydew in laboratory and field conditions and to benefit from honeydew, as it increased its longevity when honeydew was provided with water. Although no olfactory preference was observed, honeydew also stimulated oviposition by *A. mali*. The contribution of honeydew to increase the efficiency of *A. mali* as a biological control agent is discussed.

**Abstract:**

Many parasitoids need to feed on sugar sources at the adult stage. Although nectar has been proven to be a source of higher nutritional quality compared to honeydew excreted by phloem feeders, the latter can provide the necessary carbohydrates for parasitoids and increase their longevity, fecundity and host searching time. Honeydew is not only a trophic resource for parasitoids, but it can also constitute an olfactory stimulus involved in host searching. In this study, we combined longevity measurements in the laboratory, olfactometry and feeding history inference of individuals caught in the field to test the hypothesis that honeydew excreted by the aphid *Eriosoma lanigerum* could serve as a trophic resource for its parasitoid *Aphelinus mali* as well as a kairomone used by the parasitoid to discover its hosts. Results indicate that honeydew increased longevity of *A. mali* females if water was provided. Water could be necessary to feed on this food source because of its viscosity and its coating by wax. The presence of honeydew allowed longer stinging events by *A. mali* on *E. lanigerum*. However, no preference towards honeydew was observed, when given the choice. The role of honeydew excreted by *E. lanigerum* on *A. mali* feeding and searching behavior to increase its efficiency as a biological control agent is discussed.

## 1. Introduction

Generally, adult parasitoids exploit plant-derived food sources such as nectar and honeydew, whereas as larvae they feed on their insect host [1,2]. Even if certain parasitoids feed on the host haemolymph as adults, most of them supplement their diet with plant-derived food during part of their life cycle—being generally described as life-history omnivores [3]. Honeydew is often the predominant source of sugar for adult parasitoids in the agroecosystems [4,5,6], although it is generally described as being of lower nutritional quality than nectar. The nutritional value of honeydew varies among producer species [7] and could even be toxic [8], but most of the time its consumption is associated with enhanced fitness [4,9,10]. Its accessibility and ease of use may vary depending on its viscosity [9], as honeydew often crystallizes [4]. Identifying quality food sources for parasitoids is essential to optimize the design of agroecological practices and to improve their effectiveness as biological control agents [11].

Herbivore hosts and their host plants emit different cues that affect parasitoid foraging behavior [12,13,14]. These chemical signals can be volatiles emitted by the plant constitutively or induced by herbivore attacks [12,14]. Honeydew excreted by phloem feeders emits some volatile chemical cues which can be used by their parasitoids as a host-finding kairomone [15,16,17,18]. The origin of such volatiles has been presumably related to microorganisms that use honeydew as a sugar-rich substrate for their development, producing these volatiles as a by-product of the degradation of sugars and amino acids found in aphid honeydew [15,19]. These kairomones have been found to act as long-range signals detected through the parasitoid olfactory receptors [20]; moreover, honeydew produces non-volatile compounds that can act as contact host location kairomones [21,22]. There are several cues, such as epicuticular waxes on leaves, pheromones or host by-products, that are involved in food and host location that could allow the development of strategies to attract parasitoids and, consequently, enhance the biological control of pests.

*Aphelinus mali* Haldeman (Hymenoptera: Aphelinidae) is the main parasitoid of the woolly apple aphid (*Eriosoma lanigerum* Hausmann (Hemiptera: Aphididae) in apple orchards [23,24]. This aphid colonizes the canopy, trunk and roots of apple trees, causing great economical losses, and produces large amounts of powdery wax, which coats the excreted honeydew to form unsticky honeydew drops that are easily moved by nymphs. This coating of honeydew could make feeding more difficult and limit volatile emissions and host location by natural enemies. *Aphelinus mali* has been considered a host-feeding parasitoid, according to [25,26,27]. It has been observed exploiting both plant-derived food sources such as nectar and host-derived sugar sources such as honeydew. In the present study, we evaluate the role of honeydew as a food source and as a host-related kairomone for host location by assessing its effect on *A. mali* longevity, preferences and foraging behavior. We hypothesized that honeydew excreted by *E. lanigerum* feeding on apple plants should be an important sugar-rich food source for *A. mali* that may increase its longevity and act as a kairomone involved in host searching. With this purpose, we conducted a feeding experiment in controlled conditions, and to investigate the role of honeydew as a kairomone, we performed behavioral choice tests in controlled conditions. In order to demonstrate the consumption of honeydew, we evaluated the feeding patterns of *A. mali* in field conditions based on *A. mali* sugar profiles obtained with HPLC and a machine learning algorithm. We hypothesized that honeydew is one of the main food sources for *A. mali* in field conditions, especially at high aphid abundances (February in the Southern Hemisphere).

## 2. Materials and Methods

### 2.1. Insect Material

Bioassays were conducted using newly emerged *Aphelinus mali* females from *E. lanigerum* colonies collected in eight apple orchards in the Maule region (Chile). These colonies were kept in Petri dishes in the laboratory under controlled conditions (22 ± 2 °C; 60 ± 10% RH and light/dark 16:8 h photoperiod) and were cleaned of non-mummified aphids, wax and the drops of coated honeydew before parasitoid emergence started. Petri dishes were checked daily for emerged parasitoids. After parasitoid emergence, individuals were sexed and placed in clean Petri dishes with water for 24 h to mate. Parasitoids for the bioassays were naïve, less-than-two-day-old mated females and they were used only once. Honeydew was collected by placing a Petri dish under an *E. lanigerum* colony growing on its host tree and then hitting this branch. Coated honeydew was then collected using a fine brush and placed in the containers used for the assays below.

### 2.2. Longevity

Each individual was exposed to either water (*n* = 24), honeydew from *E. lanigerum* feeding on apple trees (*n* = 25), honeydew with water (*n* = 25) or no food or water (*n* = 23) in a plastic tube of 1.5 mL (hereafter: Water; Honeydew; Honeydew + Water; Control, respectively). The tubes were opened daily for 5 s to renew the air inside the plastic tube and the food and water were renewed every two days. Parasitoids were kept in the tubes until they died. Longevity was determined as the number of days that adults were alive after the emergence.

### 2.3. Infectivity of A. mali under Different Feeding Treatments

For this non-choice experiment, parasitoids were exposed to water or to three drops of coated honeydew. The experimental arena consisted of a cylinder of 1 cm in diameter and 1 cm in height with an apple branch of 1 cm in length, containing a single third or fourth instar *E. lanigerum* nymph, previously feeding on an apple tree and one of the food sources mentioned above. A single mated *A. mali* female was introduced in the arena and allowed five minutes to settle. After this settlement period, the behavior of *A. mali* in the arena was observed for 15 min under a stereomicroscope (Olympus 45X) with a homogenous cold constant light source (LED). For the analysis, two positions in the arena were considered: on the base of the arena and on the walls of the arena, and five behaviors: moving, stationary, stinging-oviposition, attacking and feeding. The behaviors “stinging-oviposition” and “attacking” were defined as in [28]. The stinging-oviposition behavior was considered to be the time that females spent with the ovipositor inserted in the body of the aphid, whereas the attacking behavior referred to the time spent exposing the ovipositor of the females with no insertion of it in the aphid’s body. Total time (sum of all the time for each of the behaviors and positions), the mean duration for each behavior and position in the arena as well as the frequency of each of the behaviors at each position were registered using the “tcltk” package of the software R v4.0.2 (The R Foundation for Statistical Computing 2020). These results allowed the calculation of the proportions of occurrences and time spent at each position in the arena for the studied behaviors. Each treatment was replicated 18 times.

### 2.4. Choice Experiment

A four-way olfactometer [29,30] was used to conduct a choice experiment with a female *A. mali.* Three different choice conditions were tested, presenting two odors in opposite sides of each branch: 1: Honeydew-Water; 2: Honeydew-Room air; 3: Water-Room air. The olfactometer was placed in a white acrylic cage illuminated with diffuse fluorescent white 18 W cold light tubes. Experiments were conducted at 22 ± 2 °C and RH 59 ± 4%. A constant airflow of 0.3 L min^−1^ was produced by a sucking PTFE (polytetrafluoroethylene) membrane pump (KNF lab, France), connected to the central opening of the olfactometer and resulting in an air flow of 75 mL min^−1^ in each branch. The resources were placed at the end of a glass Pasteur pipette: a filter paper soaked in water for the water treatment, 10 medium size drops of coated honeydew for the honeydew treatment or the control with room air. Four equal areas and a non-choice area (neutral area) were defined and drawn on a filter paper placed under the olfactometer as explained in [31]. A single female *A. mali* was introduced in the olfactometer through the central opening and its movements were recorded for 10 min using the “tcltk” package of R software (R Core Team 2012). Olfactometers were cleaned with normal dishwashing detergent and water after each assay. Time spent in each branch and in the neutral area was recorded. The time in the two opposite branches corresponding to the same odor source was summed. The olfactometer and each food source were replaced at every replicate and females were used only once. A Total of 26 (Honeydew-Room air), 24 (Honeydew-Water) and 25 (Water-Room air) parasitoids were tested in the Pettersson olfactometer.

### 2.5. Inferring Feeding History of A. mali

Using the inferential approach developed by [32], we evaluated the feeding history of *A. mali* caught in apple orchards during the months of higher abundance of *E. lanigerum* (February and March in the Southern Hemisphere). Firstly, a laboratory experiment was conducted to define the reference sugar profiles of parasitoids fed on a controlled diet. These data were then used to train a classification algorithm. The individual sugar profiles of wild parasitoids caught in the orchards were determined and the classification algorithm was applied to class each individual into 5 different classes (see below). The proportion of individuals in each class was then compared between sample dates.

#### 2.5.1. Reference Profiles

Newly emerged unfed female *A. mali* adults were collected and were individually placed in 1.5 mL plastic tubes. Females were then given one of the following feeding treatments: 2 µL of water (water), 2 µL of nectar from *Daucus carota* (Nectar), 2 µL of a 30% honey-water solution (diluted honey), 3 medium-size drops of coated honeydew excreted by *E. lanigerum* feeding on apple trees and the control without any food source (starved). Parasitoids were observed for at least three minutes and were considered to be fed when putting their mouthparts in contact with the food for more than 5 s. After feeding, parasitoids were frozen and kept at −80 °C for further HPLC analysis.

#### 2.5.2. Profiles of Field-Collected Parasitoids

Five apple orchards were selected in the Maule region of central Chile (35°26′ S 71°40′ W). The apple cultivar used was Granny Smith. Parasitoid sampling was conducted twice in February and March 2018 when populations of *E. lanigerum* were most abundant.

The parasitoid *A. mali* was collected with a manual entomological aspirator (a total of 203 adult females were collected): two people searched for the parasitoids on the whole trees (leaves, branches, trunk or aphid colonies) for 5 min. The sampling was conducted between 12:00 and 14:00 o’clock to ensure parasitoid activity. Each captured parasitoid was placed individually in a plastic 1.5 mL tube and immediately transferred to a cooler with icepacks for conservation. Once in the laboratory, parasitoids were stored at −80 °C until sugar extraction and HPLC analysis were conducted.

#### 2.5.3. Sugar Extraction for HPLC Analysis

Following the extraction method used in [6], female parasitoids were lyophilized, weighed, and crushed in plastic tubes of 1.5 mL containing glass beads using a homogenizer. The resulting powder was then diluted in 500 µL of 80% methanol-distilled water and, after vortexing, samples were placed in a water bath at 76 °C for 15 min. After the incubation period, the methanol was evaporated using a speed-vac at 42 °C (for 3 h), after which samples were diluted in 500 µL Milli-Q water. Samples were then vortexed and centrifuged (21 min; 4 °C; 14,000 rpm) to isolate the supernatant containing all soluble sugars. The supernatant was then transferred to a new tube for evaporation in the speed-vac at 42 °C (5 h). Lastly, samples were kept in a final volume of 100 µL of Milli-Q water for HPLC analysis.

A total of 10 µL of each diluted sample was injected in a DIONEX ICS 3000 system (Dionex Corp., Sunnyvale, CA, USA) equipped with a CarboPac PA1 column for HPLC analysis. The column was eluted with a 100 mM solution of NaOH and washed with 850 mM of acetate of sodium. The compartment temperature was 20 °C. Every ten samples, a standard was run to check for deviation from the calibrated values. The standards used for these analyses were glucose (2.5 mg/L), fructose (2.5 mg/L), sucrose (5 mg/L), raffinose (10 mg/L), stachyose (10 mg/L), melezitose (10 mg/L), erlose (10 mg/L) and maltose (40 mg/L). The sugar content of each individual sample was analyzed using the Chromeleon^TM^ Chomatography Data System.

### 2.6. Statistical Analysis

Raw data were checked for normality and homogeneity of variance using the Shapiro–Wilk W-test before performing the parametric test (linear model). Data were transformed to reduce heteroscedasticity for the analysis of longevity (log (x + 1)) and for the duration of stinging-oviposition and attacking behavior (sqrt). Data following a binomial distribution and showing overdispersion, such as the proportion of moving individuals, the proportion of time at the base of the arena or the proportion of times that parasitoids were at the base of the arena, a quasibinomial Generalized Linear Model was used. When data showed significant differences between food treatments, a Tukey’s HSD test was used as a post hoc test.

The inference analysis was conducted using a random forest analysis that classifies individuals into different feeding classes (unfed and fed (i.e., water, honeydew, diluted honey or nectar)). To do so, predictor variables (glucose, fructose, sucrose, GF ratio [glucose/(glucose + fructose)] and total sugars) were used and an adequate prediction method was selected using the heatmap suggested by [32]. Thus, having a ‘noise index’ (0.39) and dataset size (86), the best approach for prevalence estimation was to use a Random Forest classifier with an adjusted counting correction method. These methods were used to predict the relative frequency of parasitoids from each class (diluted honey, honeydew, nectar, starved, water) for two collection dates. The algorithm was trained and used with the help of the *randomForest* package [33]. The relative frequency of insects from each feeding class was then compared using chi-squared analysis.

All data were analyzed using the software R v4.0.2 (The R Foundation for Statistical Computing 2020).

## 3. Results

### 3.1. Longevity

The longevity of *A. mali* was significantly affected by the feeding treatments (F = 60.714; *p* < 0.001), with the starved individuals living for the shortest period of time. Parasitoids fed on honeydew-water treatment led to a far longer lifespan (Figure 1). The availability of water significantly increased the longevity of *A. mali* compared to those individuals that were exposed to honeydew alone or were starved.

### 3.2. Infectivity of A. mali under Different Feeding Treatments

Honeydew consumption influenced the mean time that *A. mali* females spend foraging surrounding their host. They were observed at the base of the arena for longer than at the walls and cover when exposed to honeydew. In addition, *A. mali* stung for almost twice as long when exposed to honeydew (F = 5.341; *p* = 0.027); however, no differences were observed for attacking (F = 1.080; *p* = 0.306) or for feeding durations (F = 0.047; *p* = 0.829) (Figure 2). Moreover, the presence of either honeydew or water did not change the frequency of stinging-oviposition (F = 0.586; *p* = 0.449) or attacking (F = 0.400; *p* = 0.532). The time spent feeding on honeydew or water was similar (F = 0.006; *p* = 0.937).

### 3.3. Choice Experiment

*Aphelinus mali* showed no preference between the provided resource combinations with honeydew (honeydew-water; honeydew-room air); however, it showed a preference toward water when given a choice between water and room air (Figure 3).

### 3.4. Inferring Feeding History of A. mali

The prediction analysis indicated that the majority of collected individuals had fed (99%) (Figure 4). In February, all the parasitoids collected fed on nectar (7%), honeydew (64%) or diluted honey (29%) (unknown sugar-rich source in apple orchards). However, in March, 2% of the parasitoid population was unfed and the fed ones were classified as fed on either water (30%) or a honey solution (68%).

## 4. Discussion

Feeding on different food sources affects the longevity of *A. mali*, as the honeydew-water treatment showed the greatest longevity compared to the other treatments. Moreover, the presence of honeydew affected the duration of foraging by this parasitoid, increasing the time spent foraging and doubling the mean time allocated to sting-oviposit its host in the presence of honeydew. However, *A. mali* does not seem to be attracted to olfactory cues from honeydew. Instead, *A. mali* seems to respond to the olfactory cues of water, such as humidity, when nothing else is offered. 

Aphid hosts are known to undergo selection pressure to minimize the nutritional benefits of honeydew for parasitoids [34] and can develop a mutualistic relationship with ants to be guarded against parasitoids in exchange for honeydew [35]. However, parasitoids have co-evolved with their aphid host to be able to utilize the resources (e.g., honeydew) that their host provides [36] and even benefit from the “enemy-free space” provided by ants [37]. *Aphelinus mali* is able to consume honeydew excreted by *E. lanigerum* [38] and some evidence suggests that there is no competition with ants such as *Lasius* spp. or *Formica cunicularia* in Europe [39]. However, interactions with the ant *Linepithema humile* has been observed with *E. lanigerum* in apple orchards in New Zealand [40]. The latter ant species is the most common ant species in central Chile; however, in our study, ants were rarely observed among *E. lanigerum* colonies in the sampled apple orchards (Personal observations). Honeydew alone showed low benefit for *A. mali*, as its longevity is lower than for those individuals that fed on water. However, if parasitoids had access to honeydew and water simultaneously, the longevity of *A. mali* was significantly greater than those individuals feeding on honeydew, water or starved. These results agree with those of [36], where authors indicated the importance of the accessibility of water together with other food sources (honey, sucrose or honeydew) to increase the longevity of parasitoids. Indeed, some studies suggest that the accessibility to water can stimulate the production of saliva that could help to dissolve the crystallized honeydew [41]. The honeydew of *E. lanigerum* is surrounded by a wax layer that may indeed hinder its consumption by *A. mali*; however, the access to water may have allowed *A. mali* to overcome this difficulty and utilize this sugar source for its benefit, increasing longevity. Moreover, the increase in relative humidity can impact on the viscosity of sugar sources such as nectar [42]; thus, the viscosity of honeydew could have decreased due to the increased humidity from the presence of water in the plastic tube and improved the accessibility of honeydew for *A. mali*.

Successful foraging for food and host is of critical importance for parasitoids to survive and reproduce [43]. Honeydew represents a low-quality sugar source for some parasitoids [4,41] and it is known to serve as a kairomone for host location [44]. In the present study, *A. mali* spent significantly more time at the base of the arena with the presence of honeydew compared to water. This behavior has been shown in other systems with parasitoids [45] where honeydew induced a longer searching time. This increased time spent on the base of the arena was not invested in feeding by *A. mali*, as it spent similar time feeding on honeydew and on water (Figure 4). In addition, honeydew stimulated the duration that *A. mali* spent in stinging-oviposition on *E. lanigerum*. Similarly, Ref. [15] found that volatiles produced by the host-associated bacteria on honeydew can act as a kairomone that attracts natural enemies and stimulates their oviposition on their host. Therefore, although honeydew alone may not be the most nutritious food source, it seems to be used as a food source for *A. mali* and as a kairomone signal for host location and oviposition stimulation, overall showing the important role of honeydew in the reproductive success of *A. mali*.

When given a choice, *A. mali* had no clear preference towards honeydew. This weak effect of honeydew on *A. mali* attractiveness may be related to the fact that the honeydew excreted by *E. lanigerum* is surrounded by a wax that could interfere with the chemical signals that pure honeydew can provide to parasitoids. Indeed, this layer of wax could avoid the release of volatiles as a result of sugar degradation by microorganisms. However, further research should be conducted to confirm the ability of this honeydew (innate or produced by microorganisms) to emit volatiles when the layer of wax breaks (e.g, when falling to the soil or the honeydew is excreted by crawlers). The latter will provide further insights into the role of honeydew coating towards parasitoid attractiveness. Furthermore, water seems to emit olfactory cues for *A. mali*, showing a significant preference when compared to room air. This result could most likely be related to the preference of this parasitoid for humid environments. *Aphelinus mali* is known to increase abundance with increasing humidity [46], which could have led to its preference towards water as a fitness gain selection. The authors in [47] have suggested that the aphid parasitoid *Aphidius rhopalosiphi* uses its sensilla as a hygroreceptor and, likewise, if *A. mali* can perceive humidity using its hygroreceptor, this may be responsible for its behavior towards water in the present study. Moreover, some parasitoids have shown a preference for humid environments, as they use humidity as a host habitat location cue [48]. The host of *A. mali*—*E. lanigerum*—absorbs the excess moisture from its food [26], which could create a humid area in the aphid colonies. This, in turn, could be an innate cue for *A. mali* to locate *E. lanigerum* colonies.

Parasitoid feeding is known to be affected by season [5,49,50] and the availability of water [36]. Thus, the proportion of sugar-fed parasitoids decreases from spring and summer to autumn, most likely due to the reduction of host and food availability in autumn compared to spring or summer. Our results confirm that parasitoids feed more from sugar resources in summer (February) compared to the beginning of autumn (March), as 32% of the parasitoids collected in March were unfed or had fed on water, whereas in February the totality of individuals had fed on sugary sources (Figure 4). However, these results are not explained by the variation of host abundance between seasons, as the greatest abundances of *E. lanigerum* populations in the sampled orchards are in February and March [51]. Temperature has been shown to influence herbivore honeydew production [52,53]. Thus, the reduction of temperatures in March may have slowed down the metabolism of the aphid and reduced honeydew production and thus its availability for *A. mali* in this month. The latter, together with the reduction of nectar sources in this period of the year, could have been responsible for the obtained results. Nevertheless, to improve our understanding of the seasonal dynamics of *A. mali* feeding in field conditions, more research is needed, increasing the number of orchards and seasons sampled. Moreover, *A. mali* predominantly feeds on an unknown food source as rich as diluted honey on both dates of sampling. Further research on the possible food sources available in apple orchards for *A. mali* is needed to reveal new approaches that promote the efficacy of this parasitoid as a biological agent of *E. lanigerum*.

## 5. Conclusions

*A. mali* is capable of utilizing honeydew excreted by its host, using water to overcome the difficulties of feeding on such a sugar source, and taking advantage by expanding its longevity. Moreover, honeydew for *A. mali* serves as a contact kairomone rather than an olfactory kairomone. Thus, *A. mali* does not utilize honeydew to search for its host at long distances, but at short distances it uses it for host acceptance and mediates host attack by stimulating oviposition. Moreover, when the availability of honeydew increases (summer) in field conditions, *A. mali* feeds on honeydew over other food sources present in the orchards.

This signifies the importance of honeydew for *A. mali* feeding, host searching and reproduction success. However, the kairomone signals that attract *A. mali* at long distances and the other food sources rich in sugars available in apple orchards are barely understood, and further research is needed to ultimately create novel approaches for *E. lanigerum* control in apple orchards.

## Figures and Tables

**Figure 1 insects-14-00426-f001:**
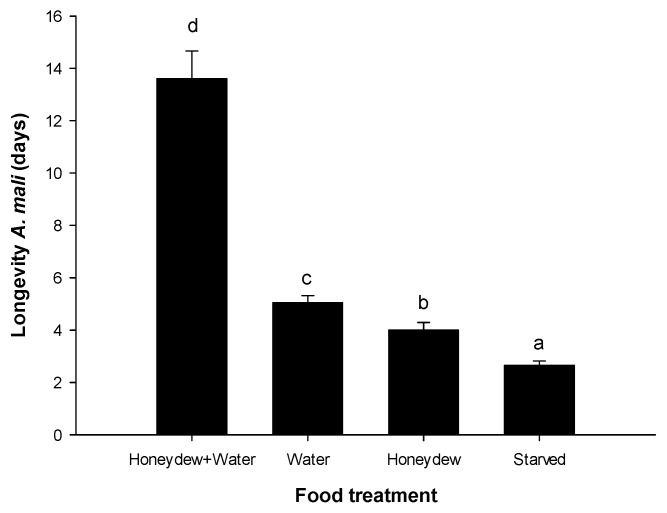
Longevity of *A. mali* either exposed to different food sources (honeydew, honeydew with water, and water) or starved. Letters above the bars indicate results of the post hoc Tukey test.

**Figure 2 insects-14-00426-f002:**
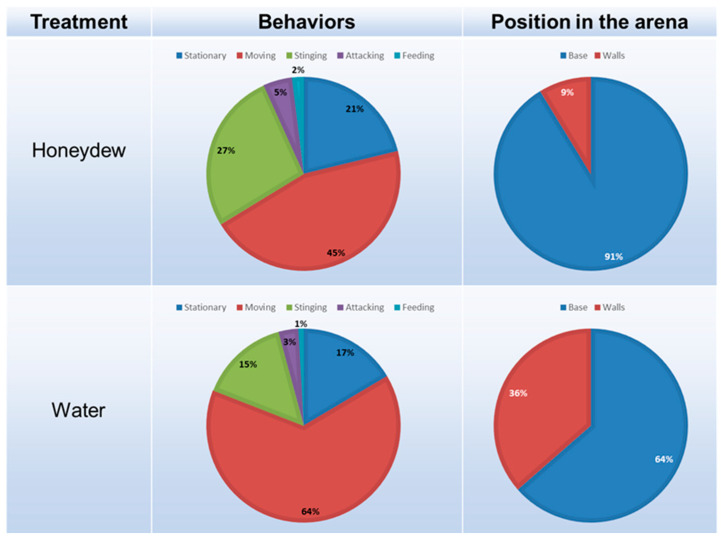
The foraging behavior of *A. mali* exposed to honeydew of *E. lanigerum* or water. The total time spent stationary, moving, stinging-oviposition, attacking or feeding as well as the total time spent at the base of the arena with the food and host sources or at the walls.

**Figure 3 insects-14-00426-f003:**
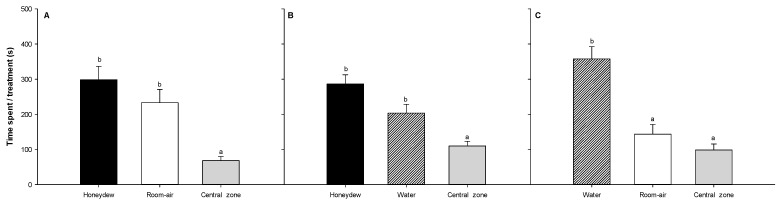
Time spent in each branch of an olfactometer by *A. mali* emerged from *E. lanigerum* when given a choice between two stimuli (Honeydew-Room air: (**A**); Honeydew-Water: (**B**); Water-Room air: (**C**)). The time spent in the central area corresponds to those parasitoids with undetermined choice [31]. Letters above the bars indicate results of the post hoc Tukey test.

**Figure 4 insects-14-00426-f004:**
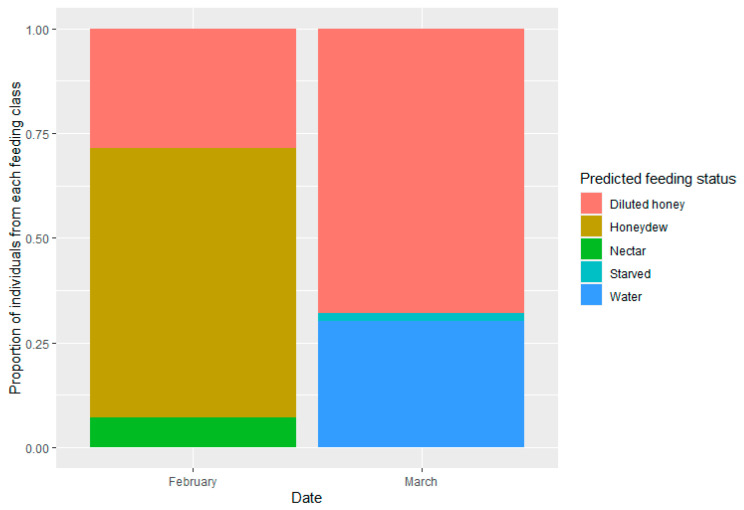
Estimated relative frequencies of *A. mali* females from each feeding class (diluted honey, honeydew, nectar, starved and water) for two sampling dates (February and March). Results were obtained using a Random Forest algorithm followed by the Adjusted Counting prevalence estimation method that classifies field-collected parasitoids.

## Data Availability

The data presented in this study are available on request from the corresponding author.

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
