# Peer review of "Honeydew Is a Food Source and a Contact Kairomone for Aphelinus mali"

_insects, 2023, doi:10.3390/insects14050426_

Round 1

Reviewer 1 Report

This manuscript by Ainara Peñalver-Cruz and colleagues is very well written and prepared. As far as lab assays go, this investigation was very well designed, replicated, and executed. The results are remarkably consistent by all measurements taken. The statistics are appropriate, and the data are clearly presented. The results are nicely placed within the context of the current, relevant literature and the conclusions are justified based on the results. This is very well-prepared manuscript, overall. This should receive quite a bit of interest from the readership of this journal and those interested in interactions between honeydew-producing sap sucking pests and their natural enemies. I recommend it for publication, but after a revision that should solve the following manuscript’s constraints:

Ants may also tend the honeydew-producing sap sucking pests protecting them from natural enemies. This article should provide details on all these fronts to provide the proper context for the work. Various honeydew-producing sap sucking pests have complex mutualistic relationships with ants which reciprocally and positively affect the partners. Honeydew-producing sap sucking pests feed on phloem sap, which is typically rich in sugars but poor in nitrogen, so they have to ingest large volumes of phloem sap and the surplus sugars are excreted as honeydew. This mutualism has benefits and costs for both partners, which are well described for many different systems; ants may provide increased colony hygiene and improved defense against natural enemies, e.g., j.biocontrol.2021.104627 and j.biocontrol.2022.105105; ACP is also producing a waxy honeydew and adding these details will improve the discussion. Ant exclusion from honeydew-producing sap sucking pests colonies, however, resulted in an increase in predation from parasitoids and naturally occurring predators, including hoverflies (see j.biocontrol.2022.105105; e.g., ACP is also producing a waxy honeydew and could be connected to this study). Also, incidence of ant-earwig antagonistic interactions was negatively correlated with earwig-aphid attack rates, interfering with woolly apple aphid biological control (see s10905-019-09722-z). Adding these details will improve the discussion in my humble opinion.

I hope you will consider revising with what I have noted in mind and resubmit. Thank you.

Author Response

Thank you for your review and comments. We have added the details that you suggested in the discussion. Please see lines 287-300. “Aphid hosts are known to undergo selection pressure to minimize the nutritional benefits of honeydew for parasitoids [34] and can develop a mutualistic relationship with ants to be guarded from parasitoids in exchange of honeydew [35]. However, parasitoids have co-evolved with their aphid host to be able to utilize the resources (e.g., honeydew) that their host provides [36] and even benefit from the “enemy-free space” provided by ants [37]. Aphelinus mali is able to consume honeydew excreted by E. lanigerum [38] and some evidence suggests that there is no competition with ants such as Lasius spp. or Formica cunicularia in Europe [39]. However, interactions with the ant Linepithema humile has been observed with E. lanigerum in apple orchards in New Zealand [40]. The latter ant species is the most common ant species in central Chile; however, in our study ants were rarely observed among E. lanigerum colonies in the sampled apple orchards (Personal observations). Honeydew alone showed low benefit for A. mali as its longevity is lower than of those individuals that fed on water.”

Reviewer 2 Report

Dear authors,

This is a very interesting work, I enjoyed reading your research. I think is well conducted, and I have just minor questions/suggestions for you:

In L78-80- I don't think this is necessary to write

L86-"whit this purpose" instead of "with this end"

in Materials and Methods:

my biggest concern is, how do you know is A. mali? you never mentioned how did you identify, did you?

How did you collect the honeydew? you didn't mention it

L-109- why the arena is so small? any reason? and why you didn't put some ventilation? instead of refreshing the air?

L-142.did you sterilize the filter paper?

L148, why detergent? and which one, you need to explain

In the olfactometer, did you change the arms after son repetitions? 

L164- you didn't mention how many repetitions

L172- again you don't mention how you identified A. mali

L287- you cannot say is "low nutritional" maybe is better to say it is not the most suitable, the same in L-132

since you mention that maybe is thanks to the hygroreceptor, the stimulus to the water, maybe you should avoid saying "olfactory"

That's all and again I think is a very nice work.

Author Response

Dear authors,

This is a very interesting work, I enjoyed reading your research. I think is well conducted, and I have just minor questions/suggestions for you:

  1. In L78-80- I don't think this is necessary to write

We thought it was interesting to add the way that this parasitoid performs host-feeding, but we agree with your suggestion since in the context of this article, this fact it is not of relevance. Therefore, we have deleted this sentence.

  1. L86-"whit this purpose" instead of "with this end"

Thank you for the suggestion, we have changed this part.

in Materials and Methods:

  1. my biggest concern is, how do you know is A. mali? you never mentioned how did you identify, did you?

Aphelinus mali is known to be the specialized hymenopterous endoparasitoid of E. lanigerum (please see Asante 1997). There are other parasitoids attacking E. lanigerum (see Asante 1997), but, there are easily differentiated to those parasitoids of the complex of Aphelinus mali, except Aphelinus niger (see Hopper et al 2012). However, the latter has been recorded only twice (Girault 1913; Gurney 1926) and both times in Australia before the introduction of A. mali in this country. Moreover, this parasitoid has been introduced for the control of E. lanigerum all over the world, including Chile where it had good establishment from the beginning of the liberations (see Howard 1929). Therefore, the probability of being mistaken in the identification of Aphelinus mali, in this case is rather improbable.  

Asante SK (1997) Natural enemies of the woolly apple aphid, Eriosoma lanigerum (Hausmann)(Hemiptera:Aphididae): a review of the world literatura. Plant Protection Quarterly 12:166-172.

Girault  AA  (1913)  Australian  Hymenoptera  Chalcidoidea  -  IV.  Memoirs  of  the  Queensland  Museum 2: 181–295.

Gurney, W.B. (1926). The woolly aphid parasite Aphelinus mali (Hald.). Agricul-tural Gazette NSW 37, 620-6.

Hopper KR, Woolley JB, Hoelmer K, Wu K, Qiao GX, Lee S (2012) An identification key to species in the mali complex of Aphelinus (Hymenoptera, Chalcidoidea) with descriptions of three new species. Joaurnal of Hymenoptera Research 26:73-96.

Howard L (1929) Aphelinus mali and its travels. Annals Entomological Society of America, 22:341-368.

  1. How did you collect the honeydew? you didn't mention it

Thank you for this question. We have added the method for collection in lines 105-107 as follow: “Honeydew was collected by placing a petri dish under a E. lanigerum colony growing on its host tree and then hitting this branch. Coated honeydew was then collected using a fine brush and placed in the containers used for the assays below.

  1. L-109- why the arena is so small? any reason? and why you didn't put some ventilation? instead of refreshing the air?

We used this arena mainly because of the size of A. mali (the body length is generally less than a mm), but also, because it is easier to manipulate and to sex this parasitoid using small plastic tubes. And regarding the ventilation, it was easier and more practical to introduce the food source without letting the parasitoid scape, if the plastic tube was with the cover. Therefore, we decided to at least refresh the air of the tube daily.

  1. L-142.did you sterilize the filter paper?

No, we didn’t sterilize the filter paper; however, the box of filter papers that we used where newly opened and we were careful on manipulating the filter paper with clean forceps.

  1. L148, why detergent? and which one, you need to explain

The detergent that we used is normal dish washing detergent and we rinse abundantly. But as the whole olfactometer is washed, there should not be any bias due to the detergent. We have specified this in line 152 of the manuscript. “Olfactometers were cleaned with normal dish washing detergent and water after each assay.

  1. In the olfactometer, did you change the arms after son repetitions? 

We changed the olfactometer and the food sources for every replication. Thus, we have specified this in the article as follow (L155-156): “The olfactometer and each food source was replaced at every replicate and females were used only once.

  1. L164- you didn't mention how many repetitions

The replications were mentioned in lines 155-156. “A Total of 26 (Honeydew-Room air), 24 (Honeydew-Water) and 25 (Water-Room air) parasitoids were tested in the Pettersson olfactometer.

  1. L172- again you don't mention how you identified A. Mali

Please see the answer to this question in the comment number 3 indicated above.

  1. L287- you cannot say is "low nutritional" maybe is better to say it is not the most suitable, the same in L-132

Thank you for this suggestion. We have rephrased this sentence as follow (L298): “Honeydew alone showed low benefit for A. mali as its longevity is lower than of those individuals that fed on water.

  1. since you mention that maybe is thanks to the hygroreceptor, the stimulus to the water, maybe you should avoid saying "olfactory"

In the line 342, we mentioned that the sensilla could be used as a hygroreceptor. Normally, the olfactory sensilla are distributed over the surface of antennae used by parasitoids to detect chemical signals from the environment. Therefore, it should be correct to mention olfactory for this case.

That's all and again I think is a very nice work.

Thank you for your kind words and your commentaries in this manuscript.

Reviewer 3 Report

This is an interesting study about food preferences of the parasitoid Aphelinus mali. Given its role as the main parasitoid of Eriosoma lanigerum, the findings of this study are important in light of future biocontrol efforts in apple orchards. I believe it is essential to dig deeper into the fine intricacies of life cycles of parasitoids we intend to use in biocontrol, as the authors did here. 

The findings of the study are presented clearly, as is the methodology used. It is a well-rounded study and I am very happy to recommend this manuscript be accepted in its current form.

Author Response

We much appreciate your words as reviewer of our manuscript, and we sincerely thank your support in accepting it.

Round 2

Reviewer 1 Report

Authors have done a stellar job addressing all of my original comments and those of other reviewers. I have no further suggestions to improve the paper. Thank you!